# Mitochondrial Transcriptome Control and Intercompartment Cross-Talk During Plant Development

**DOI:** 10.3390/cells8060583

**Published:** 2019-06-13

**Authors:** Adnan Khan Niazi, Etienne Delannoy, Rana Khalid Iqbal, Daria Mileshina, Romain Val, Marta Gabryelska, Eliza Wyszko, Ludivine Soubigou-Taconnat, Maciej Szymanski, Jan Barciszewski, Frédérique Weber-Lotfi, José Manuel Gualberto, André Dietrich

**Affiliations:** 1Institute of Plant Molecular Biology (IBMP), CNRS and University of Strasbourg, 12 rue du Général Zimmer, 67084 Strasbourg, France; adnan1753@yahoo.com (A.K.N.); rkiqbal@etu.unistra.fr (R.K.I.); d.mileshina@gmail.com (D.M.); r.val@arvalis.fr (R.V.); lotfif@unistra.fr (F.W.-L.); jose.gualberto@ibmp-cnrs.unistra.fr (J.M.G.); 2Centre of Agricultural Biochemistry and Biotechnology (CABB), University of Agriculture, Faisalabad 38000, Pakistan; 3Institute of Plant Sciences Paris-Saclay IPS2, CNRS, INRA, Université Paris-Sud, Université Evry, Université Paris-Saclay, Paris Diderot, Sorbonne Paris-Cité, 91405 Orsay, France; etienne.delannoy@inra.fr (E.D.); ludivine.soubigou-taconnat@inra.fr (L.S.-T.); 4Institute of Bioorganic Chemistry, Polish Academy of Sciences, Ul. Z. Noskowskiego 12/14, 61-704 Poznan, Poland; marta.gabryelska@gmail.com (M.G.); eliza.wyszko@ibch.poznan.pl (E.W.); Jan.Barciszewski@ibch.poznan.pl (J.B.); 5Department of Computational Biology, Institute of Molecular Biology and Biotechnology, A. Mickiewicz University Poznan, Ul. Umultowska 89, 61-614 Poznan, Poland; mszyman@amu.edu.pl; 6NanoBioMedical Centre of the Adam Mickiewicz University, Umultowska 85, 61614 Poznan, Poland

**Keywords:** anterograde regulation, cytoplasmic male sterility (CMS), plant mitochondria, retrograde regulation, ribozyme, RNA trafficking, signaling

## Abstract

We address here organellar genetic regulation and intercompartment genome coordination. We developed earlier a strategy relying on a tRNA-like shuttle to mediate import of nuclear transgene-encoded custom RNAs into mitochondria in plants. In the present work, we used this strategy to drive *trans*-cleaving hammerhead ribozymes into the organelles, to knock down specific mitochondrial RNAs and analyze the regulatory impact. In a similar approach, the tRNA mimic was used to import into mitochondria in *Arabidopsis thaliana* the *orf77*, an RNA associated with cytoplasmic male sterility in maize and possessing sequence identities with the *atp9* mitochondrial RNA. In both cases, inducible expression of the transgenes allowed to characterise early regulation and signaling responses triggered by these respective manipulations of the organellar transcriptome. The results imply that the mitochondrial transcriptome is tightly controlled by a “buffering” mechanism at the early and intermediate stages of plant development, a control that is released at later stages. On the other hand, high throughput analyses showed that knocking down a specific mitochondrial mRNA triggered a retrograde signaling and an anterograde nuclear transcriptome response involving a series of transcription factor genes and small RNAs. Our results strongly support transcriptome coordination mechanisms within the organelles and between the organelles and the nucleus.

## 1. Introduction

Mitochondria massively import proteins encoded by the nuclear genome, but still contain their own genetic system, distinct from that of the nucleo-cytosolic compartment. Mammalian mitochondria possess an extremely compact 16.5 kb circular genome with a single non-coding region that carries the promoters for transcription in both orientations [1]. In plants, mitochondrial genomes are large (usually 200–700 kb) and have a low gene density. Nevertheless, due to the occurrence of multiple promoters spread all along, they are still almost entirely transcribed, including the large non-coding regions [2]. The mitochondrial DNA (mtDNA) encodes essential subunits of the different oxidative phosphorylation (OXPHOS) complexes and one can thus presume that the mitochondrial transcriptome needs to be tightly kept under control. Studies on the regulation of mammalian mitochondrial gene expression so far highlighted a major role of post-transcriptional processes, including coordination of mitochondrial and cytosolic translation [3,4,5,6]. Translation regulation and efficiency, rather than mtDNA copy number or transcription rate, were proposed to be crucial for the synthesis of the OXPHOS subunits [7]. Methylation of the mtDNA and epigenetic-like modifications of the mitochondrial transcription factor A (TFAM) were suggested to participate in establishing the mitochondrial gene expression profiles [8]. Known nuclear gene expression regulators turned out to be also involved in mtDNA transcription [9]. Conversely, models for mtDNA transcriptional regulation by regular nuclear-encoded mitochondrial factors were proposed [1]. In plant mitochondria as well, the major genetic control points are generally considered to be post-transcriptional, although there is some transcriptional control of gene expression [4,10]. A number of processes driven by nuclear-encoded protein factors, such as 5′- and 3′-end maturation of transcripts, intron splicing and RNA editing are all potential sites of regulation [11].

As a further level of complexity, mitochondrial biogenesis and functioning involve both nuclear gene expression and mitochondrial gene expression, especially to build the protein complexes of the electron transport chain (ETC). This is likely to require a coordinated communication between the organelles and the nucleus [12]. Intercompartment coordination mechanisms include both anterograde (nucleus to organelle) and retrograde (organelle to nucleus) signals [13]. Anterograde mechanisms control organelle gene expression in response to endogenous or exogenous signals that are perceived by the nucleus. Indeed, expression of the mitochondrial genome, from DNA replication and transcription to transcript processing, editing and translation, is entirely carried out by proteins encoded by nuclear-located genes [3,14]. Retrograde regulation refers to signals sent by the organelles to communicate their functional and developmental state to the nucleus, which can then modulate anterograde control and cellular metabolism accordingly [11,15]. Mitochondrial retrograde regulation (MRR) is an important mechanism of communication between mitochondria and the nucleus and is conserved among yeasts, mammals and plants. However, the signaling molecules and the signal transduction mechanisms are diverse [15].

The present work aims to further explore these pathways with a special interest in mitochondrial genetic expression control and signaling during development. Plants offer in this respect a relevant and easily accessible working model. Establishment of the plant mitochondrial transcriptome has been documented in detail along seed germination in *A. thaliana*, but little is known about the spatial and temporal control of the organellar transcripts during post-germination development [16,17,18]. Our experiments were based on the analysis and manipulation of the organellar transcriptome in *Arabidopsis thaliana* through a novel approach set up in our laboratory and based on the existence of a natural process of transfer RNA (tRNA) import from the cytosol into mitochondria in plant cells (reviewed in [19]). Circumventing the lack of methodologies for mitochondrial transformation, we showed previously that a tRNA mimic can be used in vivo as a shuttle for importing into plant mitochondria cargo RNAs expressed from nuclear transgenes [20]. Taking *trans*-cleaving hammerhead ribozymes as cargo sequences allowed to specifically knock down mitochondrial RNAs and demonstrate that the mt-DNA-encoded MATR protein is a mitochondrial maturase involved in the splicing of multiple organellar introns [20,21]. We now used tRNA mimic-mediated shuttling of cargo RNAs to affect the mitochondrial transcriptome in *A. thaliana* plants at different developmental stages, so as to investigate the resulting regulation processes. Two strategies were applied. First, individual mitochondrial mRNAs encoding subunits of OXPHOS complexes were chosen as targets for specific *trans*-cleaving hammerhead ribozymes driven into the organelles by the tRNA mimic. In a second strategy, the tRNA mimic was used to drive into mitochondria a sequence associated with cytoplasmic male sterility (CMS). The results altogether imply that the mitochondrial transcriptome is tightly controlled by a buffering mechanism at early and intermediate stages of development. On the other hand, knocking down a major mitochondrial mRNA triggered a retrograde signaling and an anterograde response from the nucleus. Whereas it is currently considered that mitochondrial regulation processes occur essentially at the post-transcriptional stage (see above), our results support mRNA coordination mechanisms within the organelles and between the organelles and the nucleus.

## 2. Materials and Methods

### 2.1. Preparation of Gene Constructs for In Vivo Expression

For the catalytic RNA-mediated knockdown strategy, the complete sequences encoding the *trans*-ribozymes, the linker L, the *Turnip yellow mosaic virus* (TYMV) PKTLS import shuttle and the *Hepatitis delta virus* (HDV) antigenomic *cis*-cleaving ribozyme (cHDV) were assembled by polymerase chain reaction (PCR) amplification from the pCK-PSTYPKTLScHDV plasmid [20]. Direct megaprimers contained a 5’ *Xho*I site followed by the desired ribozyme and linker sequences upstream of a sequence corresponding to the 5’ region of the PKTLS. The reverse primer contained a 5’ *Spe*I site followed by a sequence annealing to the 3’ region of the cHDV [22]. The obtained PCR products were cloned into the pGEM-T vector (Promega, Madison, WI, USA) for sequencing, re-excized with the *Xho*I and *Spe*I restriction enzymes and cloned into the *Xho*I and *Spe*I sites of the inducible transcription unit (O_LexA_-46 estradiol-inducible promoter) of the pER8 vector [23], raising a series of pER8-Rzxxxx-L-PKTLS-cHDV plasmids. These were used to transform *A. thaliana* or *Nicotiana tabacum*. Sequence details can be found in Figure 1. As previously [20,21], the linker (L) to provide spacing between the *trans*-ribozyme and PKTLS motifs was selected from a pool of random sequences, based on MFOLD [24] predictions of weak RNA secondary structures unlikely to interfere either with PKTLS formation or *trans*-ribozyme binding to the target sequence.

For the CMS RNA strategy, a PCR product comprising the complete *orf77* sequence with four nucleotides of 5’-UTR and 92 nucleotides of 3’-UTR was amplified using total DNA from CMS-S maize as a template. The forward primer comprised a *Hind*III site and a sequence encoding a 6xHis tag. The reverse primer had a *Bam*HI site. The resulting product was cloned into the *Hind*III and *Bam*HI sites upstream of the sequences encoding the TYMV PKTLS and the HDV *cis*-ribozyme in an available derivative of the pUCAP plasmid [25] carrying these sequences. After sequencing of the recombinant plasmids, the assembled *orf77*-PKTLS-HDV construct was re-excized upon digestion with *Asc*I and *Pac*I and cloned into the *Asc*I and *Pac*I restriction sites of the estradiol-inducible transcription unit of the pER8 vector [23]. The resulting plasmid called pER8-*orf77*-PKTLS-cHDV was used for transformation of *A. thaliana*. For control, plants were transformed with empty pER8 plasmid. Sequence details can be found in Figure 2.

### 2.2. Nuclear Transformations

*A. thaliana* plants (ecotype Col-0) were transformed via *Agrobacterium tumefaciens* through floral dip [26]. Seeds from transformed plants were recovered and germinated on hygromycin medium under long day conditions (16 h light/8 h dark) for transformant selection. The presence and integrity of the constructs was confirmed by PCR and homozygous lines of the relevant transformants were selected.

To generate tobacco transformants, leaf discs from *N. tabacum* plants grown in vitro on germination medium (4.3 g/L DUCHEFA micro- and macroelements M0238, 10 g/L agar, 1% *w*/*v* sucrose, pH 5.7) were exposed to the relevant recombinant *A. tumefaciens* suspension and subsequently cultured in vitro on coculture medium (4.3 g/L DUCHEFA micro- and macroelements M0238, 825 mg/L NH_4_NO_3_, 2 mg/mL glycine, 100 mg/L myo-inositol, 0.5 mg/L nicotinic acid, 0.5 mg/L putrescine, 0.1 mg/L thiamine, 3% *w*/*v* sucrose, 0,8% *w*/*v* bacto-agar, 2 mg/mL benzylaminopurine (BAP), 0.05 mg/L naphthaleneacetic acid (NAA), pH 5.8) for 3–4 days at 28 °C (12 h light/12 h dark). The leaf discs were then transferred to selection medium (4.3 g/L DUCHEFA micro- and macroelements M0238, 825 mg/L NH_4_NO_3_, 2 mg/mL glycine, 100 mg/L myo-inositol, 0.5 mg/L nicotinic acid, 0.5 mg/L putrescine, 0.1 mg/L thiamine, 3% *w*/*v* sucrose, 0,8% *w*/*v* bacto-agar, 350 mg/L cefotaxime, 20 mg/L hygromycin, pH 5.8) at 25 °C (12 h light/12 h dark). The medium was changed every two to three weeks. Calli appearing on the edges were transferred to individual pots with selection medium when young leaves were formed, and planted into mineral-rich soil when roots had emerged. After acclimatization in growth chambers (25 °C, 12 h light/12 h dark) the plantlets were grown in the greenhouse. Recovered seeds were further selected through germination on hygromycin medium.

### 2.3. Seedling Growth and Trangene Induction

*A. thaliana* or *N. tabacum* transgenic seedlings were grown under long day light conditions (16h light/8h dark) or in the dark on solid agar medium. For transgene induction, the agar layer was overlaid with liquid medium containing 10 µM β-estradiol [23] Alternatively, plants grown on solid agar in the light were transferred to culture plates containing liquid medium supplemented with 10 µM ß-estradiol. Only the roots were dipping into the medium. Developmental stages considered are defined in the “Results” section. At all stages and in all growth conditions, samples of 3 to 4 entire plants were harvested each day till day 4 after induction. To avoid variations that would be due to the circadian clock, plants were always induced and collected at the same time of the day. Whole plant samples were immediately frozen in liquid nitrogen. Control plants (see the “Results” section) were grown and treated in the same way to take into account any side effects.

### 2.4. RNA Extraction, Northern Bloting and RT-qPCR Analyses

Total RNA was extracted from *A. thaliana or N. tabacum* seedlings following standard TRI Reagent protocols (Molecular Research Center). Northern blot analyses were carried out through standard protocols using non-radioactive digoxygenin-labeled probes prepared with the PCR DIG Probe Synthesis kit (Roche, Basel, Switzerland). Prior to reverse transcription, RNA samples were treated up to 3 times with RNase-free DNase 1 (Thermo Scientific, Waltham, MA, USA). Reverse transcription was carried out with SuperScript III (Invitrogen, Carlsbad, CA, USA), RevertAid (Fermentas, Waltham, MA, USA) or GoScript (Promega, Madison, WI, USA) reverse transcriptase according to the corresponding recommended protocols, using 2 pmol of a specific primer or 250 ng of a random hexanucleotide mixture. Reactions were stopped by 15 min incubation at 70 °C and the RT samples served directly for standard or real-time PCR. The absence of residual DNA contamination in the original RNA samples was assessed by the absence of PCR product amplification from cDNAs resulting from reverse transcription assays run without the enzyme.

Real time PCR (qPCR) was performed in 384-well optical plates on an iCycler (BioRad, Hercules, CA, USA) or a LightCycler 480 II (Roche) using 5 μL of PCR master mix containing 480 SYBER Green I fluorescent reporter (Roche) with 2.5 μM forward and reverse specific primers. DNA or cDNA template and water were added to a total volume of 10 μL. Each sample was performed in triplicate. Reactions were run in the following conditions: pre-heating at 95 °C for 10 min, followed by 40 cycles of 15 sec at 95 °C, 30 sec at 60 °C and 15 sec at 72 °C. Melting curves were run to assess the specificity of the products.

The nuclear genes encoding actin 2 (accession AT3G18780; AB158612), glyceraldehyde 3-phosphate dehydrogenase c (GAPDH, accession AT1G13440; AY049259) and RGS1-HXK1 interacting protein 1 (RHIP1, accession AT4G26410; BT002964), as well as the mitochondrial gene encoding mitochondrial ribosomal protein L2 (rpl2, accession BA000042, region 361051-363948) served as reference genes. To design qPCR primers, we used the Universal ProbeLibrary and the ProbeFinder software (Roche) (https://lifescience.roche.com/en_fr/articles/Universal-ProbeLibrary-System-Assay-Design.html#ProbeFinder). For short sequences that were unsuccessful with such an approach, we used the Primer 3 software [27].

RT-qPCR data obtained with RNAs from transformants expressing ribozyme-PKTLS transcripts were analyzed with the Student’s *t*-test. Statistical analyses of the mitochondrial gene expression RT-qPCR data obtained with RNAs from the *orf77*-PKTLS expressing transformants were carried out using the GraphPad Prism version 7.01 software (www.graphpad.com/). Correlation analyses, one way ANOVA, multiple comparison tests and Tukey tests were used to calculate the *p* values. *p* values <0.05 were considered statistically significant.

### 2.5. Isolation of Mitochondria and Import Assessment

To confirm that the *orf77*-PKTLS RNA expressed in the transformants was targeted to the organelles, mitochondria were isolated according to established protocols [28] from light-grown plants at day 2 after induction of transgene expression with β-estradiol. Isolated organelles were extensively treated with RNase in buffer containing 100 µg/mL RNase A and 750 U/mL RNase T1. Mitochondrial RNA was extracted following standard TRI Reagent protocols (Molecular Research Center), with 150 μL of Trizol reagent (Invitrogen) added to an amount of mitochondria equivalent to 200 μg of proteins. Total RNA was prepared in parallel from the same plants. Mitochondrial and total RNAs were probed by RT-qPCR for *orf77*-PKTLS, for specific nuclear transcripts and for selected mitochondrial transcripts, so as to characterize mitochondrial enrichment and confirm the absence of significant contamination.

### 2.6. Microarray Assays

Microarray analyses were carried out on the INRA transcriptomic platform, first at the Unité de Recherche en Génomique Végétale (URGV, Evry, France) and subsequently at the Institute of Plant Sciences Paris-Saclay (IPS2, Orsay, France), using CATMAv6.2 (Roche-NimbleGen technology, Madison, WI, USA) and CATMAv7 (AGILENT technology, Santa Clara, CA, USA) arrays. High density CATMAv6.2 microarray slides contained per chamber 270,000 primers representing all *A. thaliana* genes, i.e., 30,834 probes referring to the TAIRv8 annotation (including 476 probes of mitochondrial and chloroplast genes) (www.arabidopsis.org/), plus 1289 probes corresponding to EUGENE software predictions, 5352 probes corresponding to repeated elements, 658 probes for miRNAs, 342 probes for other non-coding RNAs (rRNAs, tRNAs, snRNAs, soRNAs) and finally 36 control probes. CATMAv7 slides contained per chamber 180,000 primers corresponding to 38,360 probes referring to the TAIRv8 annotation (including primers for mitochondrial and chloroplast genes), plus primers corresponding to EUGENE software specific predictions, primers corresponding to repeats, primers for miRNAs and other ncRNAs (rRNAs, tRNAs, snRNAs, snoRNAs) and the 36 controls. Each of these primers was designed in both orientations (forward strand in triplicate and reverse strand without duplication).

RNA samples from plants collected each day from Day 0 to Day 4 post-induction (i.e., 5 test samples and 5 control samples) were submitted to RT-qPCR analysis to assess *nad9* knockdown and individually used for microarray assays. Two to three independent biological replicates were analyzed and fluorochrome reversal (dye swap) was systematically applied. Total RNA extracted from *A. thaliana* plants following TRI-reagent protocols (Molecular Research Center) and treated twice with RNase-free DNase I (Fermentas) was finally purified using NucleoSpin RNA kits (Macherey-Nagel). Further steps were mostly adapted from Lurin et al. [29]. RNA samples were checked for quality on a bioanalyser (Agilent), quantified with a Quant-iT *RiboGreen* RNA Assay Kit (Thermo Fisher Scientific) and amplified with the Complete Whole Transcriptome Amplification kit (WTA2, Sigma-Aldrich) using random hexamer primers. After purification, 30 pmoles of cDNA per sample and per slide were hybridized overnight at 42 °C in the presence of formamide. Two micron scanning was performed with an InnoScan900 scanner (InnopsysR, Carbonne, France) and raw data were extracted using the MapixR software (InnopsysR, Carbonne, France).

### 2.7. Microarray Data Analyses

Microarray raw data comprised the logarithm of median feature pixel intensity at wavelengths 635 nm (red) and 532 nm (green). To correct the dye bias, global intensity-dependent normalization was run using the LOESS procedure [30]. Differential analysis was based on log-ratio averaging over the duplicate probes and over the technical replicates. The data were submitted to *limma* moderated t-statistics [31,32] using the R software package (www.r-project.org). No evidence that the specific variances would vary between probes was raised by *limma* and consequently the moderated t-statistics was assumed to follow a standard normal distribution. The squeezeVar function of the *limma* library was used to smooth the specific variances by computing empirical Bayes posterior means. To control the false discovery rate, adjusted *p*-values were calculated using the optimized FDR (false discovery rate) approach and the kerfdr library [33,34]. Probes with an adjusted *p*-value ≤ 0.05 were considered as differentially expressed. Complementary analyses were developed with the MapMan software (www.mapman.gabipd.org). To link the levels of the *nad9* transcript to nuclear transcripts, a sparse PLS (Partial Least Squares) regression approach was performed using the spls (v2.2) R package with the defaukt parameters, except eta = 0.5, K = 1, kappa = 0.5 [35]. The 99% confidence interval of the coefficient of each gene was calculated by bootstrapping (N = 1000). Only genes for which the confidence interval did not include 0 were considered as associated with *nad9* expression. As we have directly manipulated the *nad9* transcript levels, this association can be interpreted as a regulation of these genes by the *nad9* transcript levels. The sparse PLS analyses were performed separately for the etiolated and the light-grown seedlings using the log2 fold-changes. Microarray data from this manuscript were deposited into the CATdb database [36,37] (tools.ips2.u-psud.fr/CATdb/, Projects: RS13-02_Mitomanip and RS14-01_Mitomanip2) and into the Gene Expression Omnibus (GEO) repository [38] at the National Center for Biotechnology Information (NCBI) (accession numbers GSE127756 and GSE93122), according to the “Minimum Information About a Microarray Experiment” standards [39].

## 3. Results

### 3.1. Trans-Ribozyme-Mediated Modulation of Mitochondrial RNA Steady State Levels Depends on the Plant Developmental Stage

Four mitochondrial mRNAs, *nad9*, *sdh3*, *cox3* and *atp9*, were initially chosen as targets for specific *trans*-cleaving hammerhead ribozymes. The catalytic RNAs, *Rznad9*, *Rzsdh3*, *Rzcox3* and *Rzatp9* were designed on the same basis as in previous studies [20,21], i.e., with only 2 base pairs in helix II and a UUUU tetraloop. The *Rzatp9* ribozyme was already described before [20]. The ribozyme sequences were attached as 5’-trailors to the previously set up PKTLS mitochondrial RNA shuttle [20] through a weakly structured linker (L). The PKTLS sequence corresponded to the last 120 nucleotides of the *Turnip yellow mosaic virus* (TYMV) genomic RNA and included the tRNA-like structure and an upstream pseudoknot considered to improve the interaction with the aminoacyl-tRNA synthetase [40], which is a prerequisite for mitochondrial import [41,42]. Figure 1 shows the resulting design. The specifically designed *trans*-ribozymes associated with the linker and the PKTLS shuttle were expressed from nuclear transgenes in stably transformed *A. thaliana* plants (for cleavage of the *nad9*, *cox3* and *atp9* mitochondrial targets) and *N. tabacum* plants (for cleavage of the *sdh3* mitochondrial target). To master the expression, the sequences were placed under the control of the commonly used estradiol-inducible promoter system of the pER8 vector [23]. In this system, a chimeric transcription activator (XVE), taking together the DNA-binding domain of the bacterial repressor LexA (X), the acidic transactivating domain of the *Herpes simplex virus* VP16 factor (V) and the regulatory region of the human estrogen receptor (E), is expressed under the G10-90 strong constitutive promoter. The *trans*-activating activity of the chimeric XVE factor is triggered upon estradiol binding. The XVE-estradiol complex activates the target promoter, which consists of eight copies of the LexA operator fused upstream of a minimal 35S promoter. The activated target promoter drives the expression of the transgene, which is thus strictly dependent on estradiol delivery to the cells [23]. The XVE factor is constitutively expressed, with possible side effects. In our hands, estradiol delivery to the plants showed by itself a slight activating effect on mitochondrial genes. Also, our ribozyme-PKTLS constructs include a tRNA-like moiety that is recognized by the cognate aminoacyl-tRNA synthetase and enters tRNA pathways. Untransformed plants not exposed to estradiol were thus not appropriate controls to take into account all possible side effects and we took advantage of the fact that the SDH3 subunit is not encoded by a mitochondrial gene in *A. thaliana*, so that the *Rzsdh3* chimeric ribozyme directed against the *N. tabacum* mitochondrial *sdh3* mRNA has no RNA target in *A. thaliana* mitochondria. Transformant *A. thaliana* expressing the *Rzsdh3*-L-PKTLS chimeric ribozyme upon estradiol treatment was thus used as a control, as in previous studies [21]. Expression of the ribozyme transgenes was induced with estradiol at different developmental stages of light-grown plants and in dark-grown seedlings. Test lines and the *Rzsdh3* “no target” control line were grown, treated and harvested in parallel in strictly identical conditions. Expected knockdown of the corresponding target RNAs in test lines was analyzed every day for up to four days after onset of estradiol induction. For both the test lines and the control line, the level of the target RNA of interest at Day 1, Day 2, Day 3 and Day 4 post-induction was compared to the level at Day 0. At each growth stage, the kinetics obtained with the test lines were subsequently compared day by day to the kinetics of the *Rzsdh3* control line. Initial comparison to Day 0 and day-by-day normalization against the control at the same growth stage always gave the same profiles. *Rznad9* and *Rzatp9* were the most active in vivo and were exploited at all growth stages. In all our assays, expression of the ribozymes had no visually detectable phenotypic effects within the four-day time frame of the experiments (See for example, Appendix A).

Induced expression of the previously described *Rzatp9*-L-PKTLS RNA in *A. thaliana* seedlings at an early stage of growth (maximum 4 true leaves) was efficient (Figure 3a), but failed to trigger a significant decrease in the steady-state level of the *atp9* target mRNA within four days in a day by day analysis (Figure 3b). A more detailed kinetic analysis along 16 h of initial phase of ribozyme expression led to the same absence of significant decrease of the target RNA (Appendix A). Similarly, expression of the *Rznad9*-L-PKTLS RNA at the early stage of growth was unable to knock down the level of *nad9* mRNA in a 4-day (Figure 3c) or 16 h (Appendix A) analysis. Thus, at an early stage of development, *A. thaliana* seedlings seemed to be able to maintain the steady-state level of mRNAs like *atp9* and *nad9* in the presence of the corresponding *trans*-cleaving ribozymes. This suggests that mitochondrial function is of primary importance at that stage of growth and that a still active transcription might efficiently “buffer” transcriptome changes through replacement of cleaved RNAs.

At an intermediate stage of *A. thaliana* development (up to 10 true leaves), expression (Figure 3d) and mitochondrial import of the *Rznad9*-L-PKTLS RNA resulted in a striking “bounce-back” profile of the *nad9* transcript level. Figure 3e shows that, following initial efficient knockdown of the *nad9* mRNA at Day 1 after onset of *Rznad9*-L-PKTLS expression, the steady-state level of the target RNA went back to normal at Day 2, before dropping again at Day 3 and 4. The profile established by RT-qPCR was confirmed by northern blot (Appendix A). The *atp9* transcript level also showed a marked bounce-back profile upon expression and mitochondrial import of the *Rzatp9*-L-PKTLS RNA at the intermediate stage of *A. thaliana* development (Figure 3f). Further analyses showed a similar bounce-back kinetics for the steady-state level of the *cox3* mitochondrial mRNA (Figure 3h) upon induced expression of the corresponding *Rzcox3*-L-PKTLS ribozyme in light-grown *A. thaliana* plants at the intermediate stage of development (Figure 3g). Thus, the bounce-back profile of the RNA steady-state level was not restricted to a given target, but appeared to be potentially a general response of mitochondrial RNAs to the expression and organellar import of specific ribozymes at intermediate stages of growth. Moreover, the phenomenon was not either restricted to *A. thaliana*, as the mitochondrial *sdh3* mRNA level responded in the same way to the expression of the *Rzsdh3*-L-PKTLS RNA in *N. tabacum* plants at the equivalent developmental stage (Figure 3i,j). These observations altogether imply that the buffering mechanism that might operate at the early stage of development (Figure 3b,c and Appendix A) becomes looser at a later stage, but disturbance of the mitochondrial transcriptome through the knockdown of an individual mitochondrial RNA is still sensed and transiently complemented. Notably, ribozyme-mediated knockdown of the mitochondrial *matR* mRNA encoding an organellar splicing factor was previously shown to be efficient and continuous in *A. thaliana* plants at an intermediate developmental stage [21]. Transcriptomic buffering might thus mainly apply to mRNAs encoding subunits of the OXPHOS complexes.

Continuous knockdown of the *nad9* target RNA (Figure 4b), or of the *atp9* target RNA (Figure 4c) was recorded upon *Rznad9*-L-PKTLS (Figure 4a) or *Rzatp9*-L-PKTLS expression in *A. thaliana* plants at the bolting stage of development (up to 18 true leaves). Mitochondrial transcriptome surveillance and buffering thus seems to be progressively released at later developmental stages. Notably, continuous and strong knockdown of the *atp9* mRNA was also obtained earlier in flowering plants expressing the *Rzatp9*-L-PKTLS RNA [20].

Remarkably, a bounce-back profile of the *nad9* mRNA level (Figure 4e) was also observed when expressing the *Rznad9*-L-PKTLS transcript in 10-day-old *A. thaliana* seedlings grown in the dark (Figure 4d), indicating that the phenomenon was related to the age and importance of the mitochondrial function rather than to the growth conditions *per se*.

### 3.2. CMS RNA-Triggered Modulation of Mitochondrial RNA Steady State Levels Depends on the Plant Developmental Stage

The possibility that physiological conditions might impair ribozyme cleavage activity at some developmental stages cannot be completely ruled out. In particular, even when optimized, ribozymes require a minimal free Mg^++^ concentration. We thus developed a parallel strategy to affect the mitochondrial transcriptome and test its reactivity. The concept was to use the PKTLS shuttle to drive a CMS RNA into the mitochondria of *A. thaliana* plants. Mitochondria of maize lines with the CMS-S cytoplasm express characteristic transcripts that carry a specific open reading frame called *orf77* ([43] and references therein). The *orf77* is a chimera combining sequences of unknown origin with sequences derived from the mitochondrial *atp9* gene (Appendix A). Due to these similarities, it is hypothesized that the *orf77* transcript can impair *atp9* expression at the RNA or protein level. ATP9 is a core component of the F_0_ moiety of the mitochondrial ATP synthase located at the inner membrane. Ten or more ATP9 subunits assemble to form an inner membrane-embedded, hydrophobic ring that functions as a rotor pumping protons from the intermembrane space to the matrix side. ATP9 is thus a major protein for mitochondrial function.

In the present work, we imported the maize *orf77* CMS-S RNA into mitochondria in *A. thaliana* plants, so as to analyze the resulting impact on the whole mitochondrial transcriptome. Similarly, to the above *trans*-cleaving ribozymes, the *orf77* sequence was attached to the PKTLS shuttle as a 5’-trailor (Figure 2) and placed under the control of the estradiol-inducible promoter system in the pER8 expression vector [23]. The *orf77*-PKTLS RNA was expressed from a nuclear transgene in stably transformed *A. thaliana* plants (Appendix A). Expression was induced with estradiol at different developmental stages of light-grown plants and in dark-grown plants. In this strategy, plants transformed with a pER8 plasmid deprived of transgene served as a control. The resulting impact on the whole mitochondrial transcriptome was analyzed day by day for up to four days after onset of estradiol induction. At all stages, *orf77*-PKTLS expression had no visually detectable phenotypic effects on the plants within this time frame (Figure 5 and Appendix A). In this case, the cargo sequence attached to the PKTLS was longer and we could not rely on a well-defined activity to confirm organellar targeting. Purified, RNase-treated mitochondria were thus prepared from *A. thaliana* plants expressing the transgene and organellar import of the *orf77*-PKTLS transcript was confirmed by RT-qPCR probing, which showed about tenfold enrichment of the *orf77*-PKTLS RNA in the mitochondrial fraction *versus* the actin and GAPDH nuclear transcripts (Appendix A).

As for ribozymes, the effect of *orf77*-PKTLS expression and mitochondrial import on the levels of essentially all mitochondrial RNAs was characterized in transformed *A. thaliana* plants at different growth stages in the light and in dark-grown seedlings. A synthetic scheme of the results is given in Figure 5, while detailed data are compiled in Table 1. The analyses confirmed the transcriptome buffering process highlighted above in the case of *trans*-ribozyme-mediated RNA knockdown. Induced expression of the *orf77*-PKTLS RNA in transgenic *A. thaliana* seedlings at an early stage of development (maximum 4 true leaves) had almost no effect on the mitochondrial transcriptome, with only 3 RNAs positively or negatively affected and only at Day 2. At an intermediate stage of development, i.e., the stage where ribozyme-mediated target knockdown showed a bounce-back profile (Figure 3e,f,h,j), transcriptome buffering became somehow looser and up to eight mitochondrial RNAs were affected as a whole. Finally, at the bolting stage buffering was essentially released, as up to 27 mitochondrial RNAs were positively or negatively affected in response to *orf77*-PKTLS expression and organellar import. Expressing the *orf77*-PKTLS RNA in dark-grown seedlings affected up to 17 mitochondrial RNAs.

As mentioned, *orf77* contains sequences from the *atp9* gene (Appendix A), so that the *orf77*-PKTLS transcript was expected to interfere with the *atp9* mRNA. Remarkably, such an effect was indeed observed, but only at the bolting stage, i.e., at a stage where the transcriptome buffering was released. The level of *atp9* decreased at Day 2 and Day 4, while increasing at Day 3 (Table 1), further suggesting a dynamic process. Like *atp9*, *mttB* and *rpl5* showed some modulation at the bolting stage, with an initial decrease and subsequent increase at Day 4.

Except in early stage light-grown seedlings, all genetic functions encoded by the mtDNA, i.e., OXPHOS subunits, biogenesis co-factors (especially cytochrome c maturation factors), as well as ribosomal proteins and rRNAs, were represented in the organellar transcriptome response to *orf77*-PKTLS expression and mitochondrial import. Strikingly, the mitochondrial genes affected at the bolting stage were essentially downregulated, while the limited set of reactive genes at the intermediate stage was upregulated, with the exception of the cytochrome c maturation gene *ccmFN2* (Table 1). It is tempting to speculate that upregulation at the intermediate stage might be representative for transcriptome buffering, as proposed above, while general downregulation at the bolting stage would reflect the release of such a compensation process. Due to these differences, *nad5b*, *cob*, *atp4*, *rpl2*, *rpl5* and *rps7* showed an opposite behaviour between the intermediate and bolting stages. For instance, *atp4* was upregulated by about 50% at the intermediate stage and downregulated by 50% at the bolting stage.

The response to *orf77*-PKTLS expression and mitochondrial import in 10-day-old dark-grown seedlings was more balanced between up- and downregulation. As for light-grown intermediate stage samples, this led to an opposite behaviour of a number of transcripts, like for example the different *nad* mRNAs or *cox1*, *versus* the bolting stage of light-grown plants (Table 1). Notably, as intermediate stage light-grown samples, dark-grown seedlings also showed a bounce-back profile in ribozyme-mediated knockdown assays (see above, § *3.1.* and Figure 4e). When affected, *ccmFN2* was always downregulated, whichever the growth conditions or growth stage (Table 1).

### 3.3. Ribozyme-Mediated Knockdown of a Mitochondrial mRNA Triggers a Nuclear Transcriptome Response

Microarray analyses revealed that continuous ribozyme-directed knockdown of the *nad9* mitochondrial mRNA in light-grown *A. thaliana* plants at the bolting stage of development affected the level of multiple nuclear-encoded RNAs. However, the extent of *nad9* knockdown differed from one kinetics to the other for the samples from biological replicates dedicated to the microarray assays, so that we decided to perform a sparse PLS analysis of the data (Appendix A). The latter highlighted the significant regulation of 208 genes by the ribozyme-directed knockdown of *nad9* (Appendix A). Notably, these included as many as 12 upregulated (Table 2) and 7 downregulated (Table 3) transcription factor genes, as well as 12 upregulated (Table 2) and 8 downregulated (Table 3) small nucleolar RNA (snoRNA) genes. Downregulation of the questioned miR414 microRNA was also noticed (Table 3), as well as up- or downregulation of genes involved in signaling, RNA metabolism or translation (Table 2 and Table 3). Successful knockdown of *nad9*, i.e., a major mitochondrial mRNA encoding an OXPHOS subunit, in light-grown mature plants thus appeared to trigger a complex nuclear response driven by retrograde/anterograde regulation processes.

To cast off the effects of photosynthesis processes and chloroplast pathways, and try to simplify the picture, microarray assays were continued with RNAs from 10-day-old dark-grown *A. thaliana* seedlings expressing the *Rznad9*-L-PKTLS chimeric ribozyme. In this case, the mitochondrial transcriptome was constrained by buffering, so that the response was also expected to be representative for the moderate bounce-back profile of *nad9* knockdown in dark-grown plants (see Figure 4e). Microarray analyses of the RNAs from dark-grown *A. thaliana* seedlings expressing the *Rznad9*-L-PKTLS ribozyme (Appendix A) revealed that the number of differentially expressed genes reached 3046 at Day 1 and then dropped to 1622 and 416 at Day 2 and Day 3, before re-increasing to 959 at Day 4. Sparse PLS sorting of these data (Appendix A) showed a regulation of 49 genes by the ribozyme-directed knockdown of *nad9* (Appendix A), including 3 downregulated transcription factor genes, as well as genes involved in signaling or RNA metabolism (Table 4). In parallel, we mined the original microarray data for modulation of transcripts encoding proteins shown or predicted to localize to mitochondria (Appendix A). Interestingly, the analysis pointed to upregulation of a series of members of the large “small auxin upregulated RNAs” (SAUR) family, as well as to upregulation of transcripts for cytokinin and auxin response factors (Table 5), suggesting a hormone signaling-related process. MapMan analysis of the data also pointed to hormone metabolism pathways and especially to auxin-regulated genes. Conversely, downregulation of a jasmonate-responding factor, of a set of pentatricopeptide repeat (PPR) proteins and of further RNA metabolism factors was noticed (Table 5). Also, to be noted, the knockdown of *nad9* in dark-grown *A. thaliana* seedlings negatively affected the level of a series of mitochondrial mRNAs encoding subunits of OXPHOS complexes or ribosomal proteins, while mRNAs coding for biogenesis co-factors like CCM factors or MATR were little affected (Appendix A).

## 4. Discussion

All complexes of the mitochondrial OXPHOS chain consist of subunits encoded by the nuclear genome and subunits expressed from mitochondrial genes, hence the idea of a coordinated expression of the two genomes. Nevertheless, according to the available data mtDNA transcription and mitochondrial transcript levels did not seem to be major regulation issues by themselves, so that mitochondrial genetic control is considered to occur mainly at the post-transcriptional level [3,4]. However, over the past decades, there have been a number of reports describing changes in the abundance of specific mitochondrial transcripts during plant growth [45], such as during the development of wheat leaves [46] or seed germination in maize and rice [47,48,49]. Variations in the steady state levels of a number of mitochondrial mRNAs were also observed with *A. thaliana* nuclear mutants deficient for the editing or splicing of a given organellar RNA [50,51,52,53]. The question of a control of mitochondrial gene expression at the transcriptome level, connected with mitochondrial retrograde pathways, can thus still be raised. So-called intergenomic signaling has been characterized in yeast [54]. Unaffected by respiration and metabolic signals, the process requires mtDNA and appears as a direct communication between the mitochondrial and nuclear genetic systems. Similarly, the idea of an involvement of chloroplast transcription in signaling to the nucleus has been put forward [55,56].

The present work follows this line. We disturbed the mitochondrial transcriptome in plants at different stages of development through specific in vivo knockdown of different organellar target mRNAs or introduction of a CMS RNA into mitochondria. The ribozymes and the CMS RNA were expressed from an inducible promoter, so that at each growth stage the plants were in a regular physiological configuration before induction. This enabled to reveal early and specific events, a notable advantage over the use of mutants that would have to cope with the mutation and adapt permanently their pathways. Also, to be noted, transgene expression upon single estradiol induction is transient and lasts only a few days. The question of the existence of transcriptome control within mitochondria and of intergenic coordination could then be addressed by analyzing the mitochondrial and nuclear responses. Remarkably, the observations highlighted RNA control mechanisms in mitochondria with relation to the plant developmental stage. The mitochondrial transcriptome appeared to be tightly constrained in young, actively developing seedlings, as ribozyme-directed target RNA knockdown was not observed at that stage and the *orf77* CMS RNA triggered almost no effect. This suggests in particular that in young seedlings high mitochondrial transcription rates [57] might compensate negative variations of RNA levels and “protect” the transcriptome at a growth stage where efficient mitochondrial activity is essential. A corollary of such a hypothesis would be that changes affecting the organellar transcriptome can be “sensed”. Compensation became only transient at an intermediate stage of development, perhaps due to limited RNA synthesis, hence the bounce-back effect. Finally, at the bolting stage, ribozyme-mediated knockdown of target RNAs became efficient. Both *nad9* knockdown and organellar targeting of the *orf77* CMS RNA (presumed competitor of *atp9*) were then associated with a decrease of many mitochondrial RNAs, especially of mRNAs encoding subunits of OXPHOS complexes, suggesting a coordination and cohesion of the mitochondrial transcriptome. According to complementary analyses, the observed decrease of mitochondrial RNAs was not associated with variations in mtDNA copy number or in the number of mitochondria per cell. Notably, directed knockdown of *nad9*, i.e., a mitochondrial mRNA encoding a subunit of an OXPHOS complex, negatively affected the level of many other mitochondrial mRNAs encoding OXPHOS subunits or ribosome subunits. By contrast, in earlier experiments knockdown of the *matR* mitochondrial mRNA, encoding a maturase, affected intron splicing, but had limited effect on the level of intronless mitochondrial RNAs [21], suggesting that the mitochondrial transcriptome might be somehow coordinated to optimize expression of OXPHOS complex subunits.

As revealed by microarray analyses, successful ribozyme-mediated knockdown of *nad9*, i.e., a mitochondrial mRNA encoding a subunit of an OXPHOS complex, positively or negatively affected the level of multiple nuclear-encoded RNAs. This implies that directed alteration of the mitochondrial transcriptome resulted in a retrograde/anterograde response. The effect was fast, starting right away from the first day of ribozyme expression and knockdown of the target RNA, which implies a very fast retrograde signaling and raises the question of the nature of such a signaling. So far described retrograde signals are essentially metabolic signals sent to inform the nucleo-cytosolic compartment about the status of the organelles or resulting from impaired organellar function or biogenesis [54,58,59,60]. In particular, different signaling pathways respond to OXPHOS chain dysfunction. Our earlier studies showed that the profile of the mitochondrial OXPHOS complexes was still unaffected in plant cells four days after onset of ribozyme-mediated knockdown of the *atp9* mRNA [20]. Also, the level of MATR protein started to decrease significantly only from day 4 after onset of ribozyme-mediated knockdown of the *matR* mRNA [21]. With such a slow turnover of mitochondrial proteins, it seems unlikely that mitochondrial respiratory and metabolic dysfunction would occur fast enough to generate the signals underlying the observed retrograde response to the knockdown of the *nad9* mRNA. The data presented here thus support the idea of a fast communication between the mitochondrial and nuclear genomes through yet to be identified signals. Mitochondrial-nuclear communication through intergenomic signaling described in yeast was suggested to involve the Abf1p transcription factor and seemed indeed to be distinct from regulation activated by respiration dysfunction [54]. On the other hand, it was shown that sigma factors utilized by chloroplast-encoded RNA polymerase to transcribe specific sets of plastid genes are a source of retrograde signals to the nucleus [56]. However, no significant up- or downregulation of the nuclear genes for mitochondrial RNA polymerases ever appeared in our RT-qPCR and microarray analyses in response to *nad9* knockdown. It can still be hypothesized that other regular nuclear-encoded factors involved in plant mitochondrial transcription play a regulatory role.

Whichever the mitochondrial signal, ribozyme-mediated *nad9* knockdown triggered an anterograde response involving a series of nuclear transcription factors, either in light-grown or dark-grown seedlings. A number of these belonged to families highlighted in previous studies as taking part in mitochondrion-nucleus communication during plant growth, development, defense or stress (MYB, WRKY, ARR, AP2/EREBP, IAA, bHLH, ANAC/NAM) [14,61,62]. Many of them are responsive to hormone signaling. Notably, a meta-analysis of available datasets identified the plant hormones abscissic acid, auxin, cytokinin, jasmonate, and salicylic acid as main regulators of mitochondrial functions [61]. Modulation of the miR414 micro RNA (miRNA) level in response to *nad9* knockdown was also in line. miR414 was proposed earlier to be involved in the regulation of the genes participating in auxin response pathways and energy metabolism during plant embryonic development [63] and a number of its target genes were reported to play an important role in hormone signal transduction, especially in the auxin and ethylene pathways [64]. miR164 is a further miRNA highlighted in previous mitochondrion-nucleus communication analyses [61]. Also striking is the up- or downregulation of a number of small nucleolar RNAs (snoRNAs) in response to *nad9* knockdown. Beyond their fundamental role in the modification and processing of ribosomal RNAs, snoRNAs recently appeared to be a source of short regulatory RNAs [65,66].

Mining the data to identify proteins potentially sent to the mitochondria upon *nad9* knockdown in dark-grown seedlings in turn highlighted some interesting features. Firstly, the results suggest an upregulation of putative mitochondrial auxin-responsive proteins encoded by small auxin upregulated RNAs (SAUR), as well as further hormone-responsive factors (Table 5). The family of SAUR auxin-responsive genes has about 60–140 members in most higher plant species [67] and plays a central role in fine-tuning of growth in response to auxin or other hormones. Whether the SAUR proteins pointed out in our analyses are indeed targeted into mitochondria and have a specific functional or regulatory activity in there remains to be investigated in the future. The second remarkable feature highlighted by the analyses was a downregulation of a number of putative mitochondrial RNA metabolism factors, especially pentatricopeptide repeat (PPR) proteins (Table 5). PPR proteins in plants are encoded by a large family of nuclear genes with up to 800 members and most of them are targeted to the organelles [68,69,70]. Aiding in RNA editing, maturation, stabilisation, intron splicing, transcription and translation of organellar genes, they are major RNA metabolism factors in both mitochondria and chloroplasts. The AT4G01030 PPR gene highlighted in our plant study was annotated as coding for an editing factor, while endonuclease activity was attributed to the protein encoded by the AT4G21190 gene. The function of the remaining ones (Table 5) remains to be investigated and experimental evidence is needed to confirm mitochondrial targeting.

## 5. Conclusions

In conclusion, establishing directed manipulation of the mitochondrial transcriptome brought evidence that RNA levels are controlled and coordinated in plant mitochondria, and that interfering with specific mitochondrial transcripts causes a retrograde response that affects the nuclear transcriptome and triggers an anterograde response. The exact mechanisms underlying the observed transcriptome control and intercompartment cross-talk are still to be deciphered, but they do not seem to rely on signals that would be generated by mitochondrial metabolic dysfunction. While the control of mitochondrial gene expression had so far mostly been documented at the post-transcriptional level, our observations introduce a broader and more integrated view of the genetic regulation in the organelles.

## 6. Patents

Commercial use of the ribozyme/PKTLS strategy to manipulate mitochondrial transcripts is covered by patent Number WO 2010/031918.

## Figures and Tables

**Figure 1 cells-08-00583-f001:**
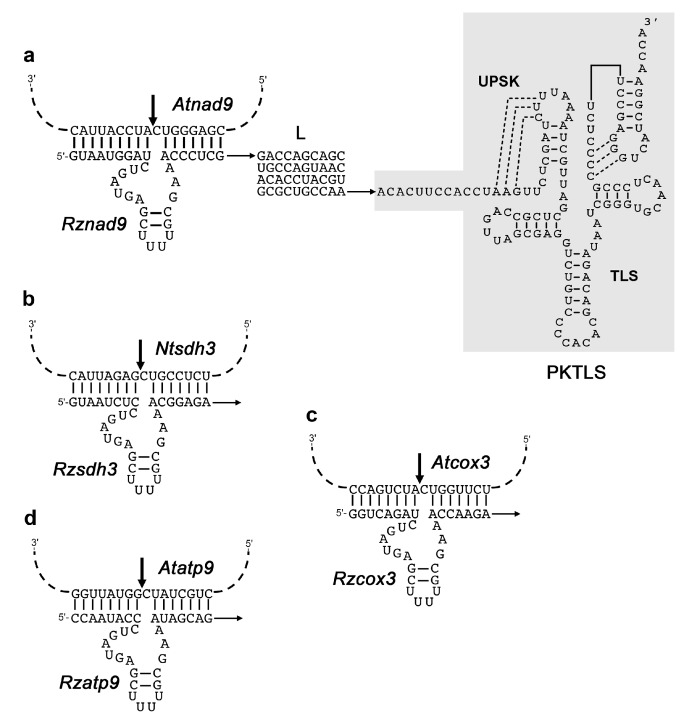
Structure of the ribozyme-PKTLS chimeric RNAs targeted to mitochondria in the present study. (**a**) Structure of the *Rznad9*-L-PKTLS RNA. The *trans*-cleaving ribozyme *Rznad9* directed against the *A. thaliana* mitochondrial *nad9* mRNA (*Atnad9*) is attached to the 5’-end of the PKTLS shuttle via a 40 nucleotide linker (L) selected from a pool of random sequences. The *Rznad9* hammerhead sequence is annealed to its target sequence motif in the *Atnad9* mRNA. The *Rzsdh3* (**b**), *Rzcox3* (**c**) and *Rzatp9* (**d**) ribozymes, directed against the *N. tabacum sdh3* (*Ntsdh3*) mRNA, the *A. thaliana cox3* (*Atcox3*) mRNA and the *A. thaliana atp9* (*Atatp9*) mRNA are attached to the same L-PKTLS moiety to generate the *Rzsdh3*-L-PKTLS, *Rzcox3*-L-PKTLS and *Rzatp9*-L-PKTLS, respectively. Ribozyme cleavage sites are indicated in the target sequence motifs by thick arrows. Their precise location is after position 421 in the *A. thaliana nad9* coding sequence (24662-25234 in accession JF729201), after position 216 in the *N. tabacum sdh3* coding sequence (77198-77524 in accession BA000042, complementary strand), after position 685 in the *A. thaliana cox3* coding sequence (328926-329723 in accession JF729201, complementary strand), and after position 99 in the *A. thaliana atp9* coding sequence (269920-270177 in accession JF729201, complementary strand). The *Rzsdh3* and *Rzatp9* ribozymes are as described earlier in Sultan et al. [21] and Val et al. [20], respectively.

**Figure 2 cells-08-00583-f002:**
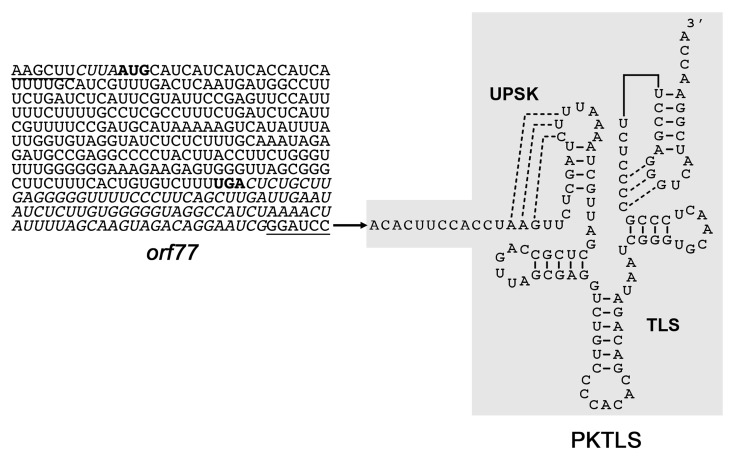
Structure of the *orf77*-PKTLS RNA targeted to mitochondria in the present study. The *Zea mays* CMS-S-specific *orf77* coding sequence (initiation codon and termination codon in bold) with its four nucleotide upstream and 92 nucleotide downstream sequences (italics) is directly attached to the 5’-end of the PKTLS shuttle. Short additional sequences deriving from the *Hind*III and *Bam*HI restriction sites introduced for cloning purposes are underlined. The chimeric RNA was expressed from a nuclear transgene and driven into the organelles by the PKTLS moiety.

**Figure 3 cells-08-00583-f003:**
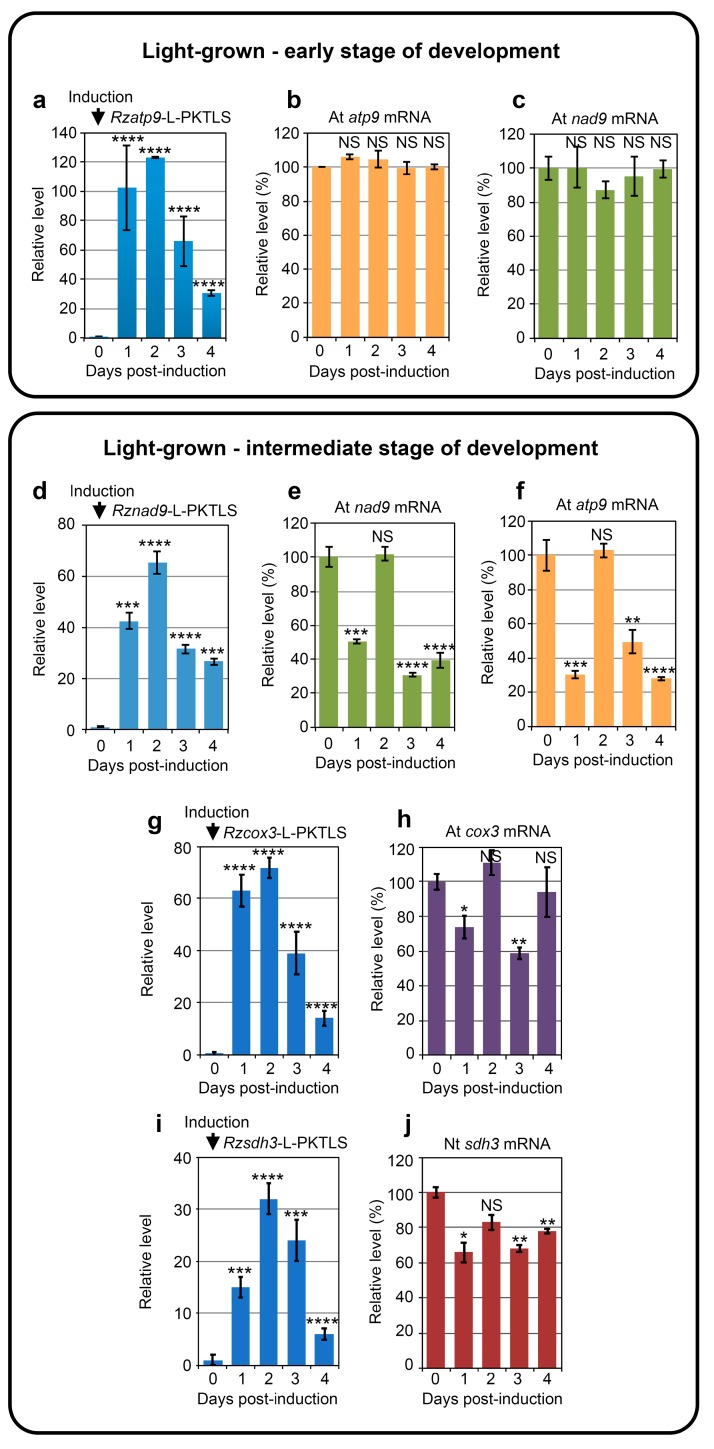
Chimeric ribozyme expression and knockdown of steady-state levels of mitochondrial target RNAs in transformed seedlings at different stages of growth. (**a**–**h**) *A. thaliana* control seeds and seeds carrying the *Rzatp9*-L-PKTLS, *Rznad9*-L-PKTLS or *Rzcox3*-L-PKTLS transgene were germinated in the light on solid MS-agar medium. Plants at early stage (upper panel) or intermediate stage (lower panel) of development were transferred at Day 0 to wells in culture plates containing liquid medium supplemented with estradiol for transgene induction. Kinetics of induced expression of the *Rzatp9*-L-PKTLS (**a**), *Rznad9*-L-PKTLS (**d**) or *Rzcox3*-L-PKTLS (**g**) RNA and of the steady-state level of the mitochondrial *atp9* (**b**,**f**), *nad9* (**c**,**e**) or *cox3* (**h**) target RNA were analyzed by RT-qPCR with total RNA from plant samples collected each day from Day 0 to Day 4 post-induction. (**i**,**j**) Transformant *N. tabacum* carrying the *Rzsdh3*-L-PKTLS transgene was germinated in the light on solid MS-agar medium and transferred at intermediate stage of development to liquid medium supplemented with estradiol for transgene induction. Kinetics of induced expression of the *Rzsdh3*-L-PKTLS RNA (**i**) and of the steady-state level of the mitochondrial *sdh3* target RNA (**j**) were analyzed by RT-qPCR with total RNA from transformed plant samples collected each day from Day 0 to Day 4 post-induction. Data from three independent biological replicates were analyzed with the Student’s *t*-test; NS = not significant; * = *p* ≤ 0.05; ** = *p* ≤ 0.01; *** = *p* ≤ 0.001; **** = *p* ≤ 0.0001.

**Figure 4 cells-08-00583-f004:**
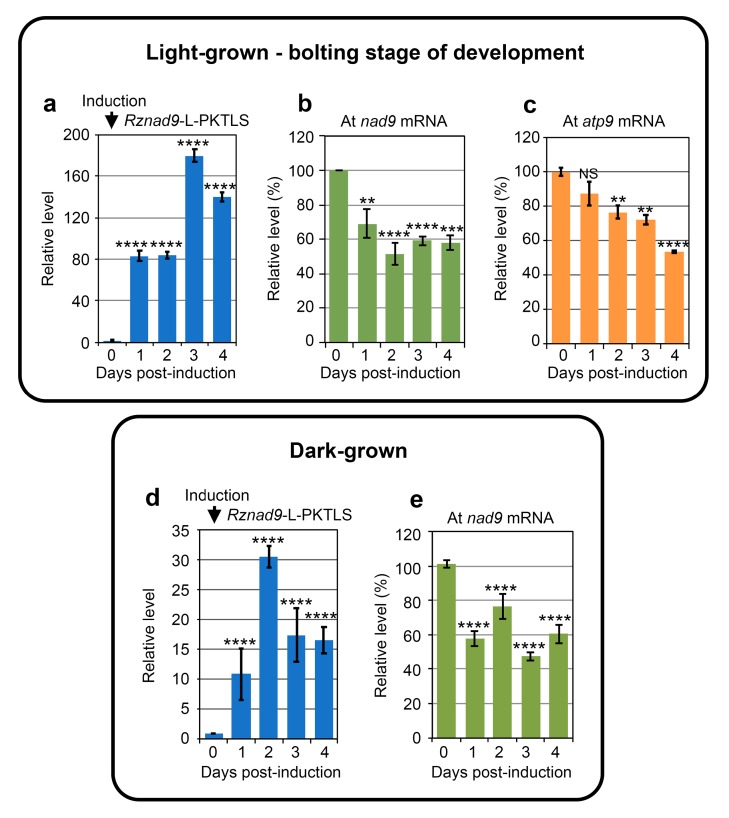
Chimeric ribozyme expression and knockdown of steady-state levels of mitochondrial target RNAs in transformed seedlings in different conditions. (**a**–**e**) *A. thaliana* control seedlings and seedlings carrying the *Rznad9*-L-PKTLS, or the *Rzatp9*-L-PKTLS transgene were grown in the light (upper panel) or in the dark (lower panel) on solid MS-agar medium. Plants at bolting stage of development (upper panel) were transferred at Day 0 to wells in culture plates containing liquid medium supplemented with estradiol for transgene induction. Kinetics of induced expression of the *Rznad9*-L-PKTLS RNA (**a**), of the steady-state level of the mitochondrial *nad9* target RNA (**b**) and of the steady-state level of the mitochondrial *atp9* target RNA (**c**) were analyzed by RT-qPCR with total RNA from plant samples collected each day from Day 0 to Day 4 post-induction. Plates with ten-day-old seedlings grown in the dark (lower panel) were overlayed with liquid medium supplemented with estradiol for transgene induction. Kinetics of induced expression of the *Rznad9*-L-PKTLS RNA (**d**) and of the steady-state level of the mitochondrial *nad9* target RNA (**e**) were analyzed by RT-qPCR with total RNA from plant samples collected each day from Day 0 to Day 4 post-induction. Data from three independent biological replicates were analyzed with the Student’s *t*-test; NS = not significant; * = *p* ≤ 0.05; ** = *p* ≤ 0.01; *** = *p* ≤ 0.001; **** = *p* ≤ 0.0001.

**Figure 5 cells-08-00583-f005:**
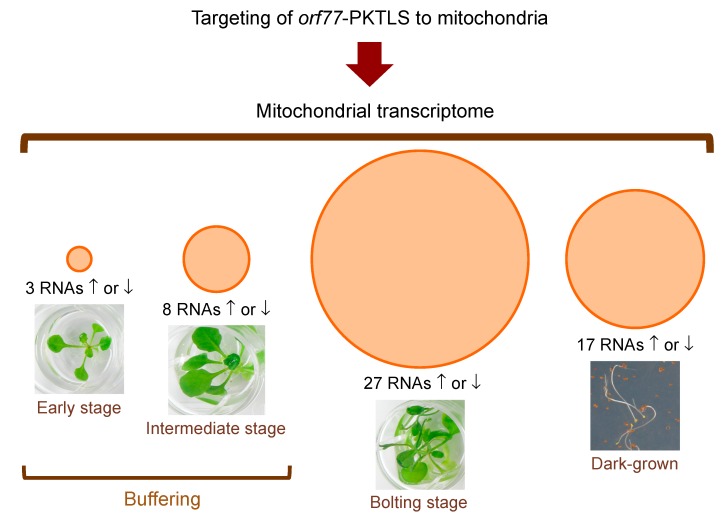
Synthetic scheme representing the impact of *orf77*-PKTLS expression on the mitochondrial transcriptome in transformed *A. thaliana* plants at different stages of growth in the light and at 10 days of growth in the dark. Plants were grown to the appropriate stage and the transgene was induced with estradiol as in Figure 3 and Figure 4. Following transgene induction, samples were collected each day from Day 0 to Day 4 post-induction and RNAs were analyzed by RT-qPCR. The total numbers of positively or negatively affected transcripts combining all daily samples for a given growth stage are indicated and represented by proportional circular areas. Detailed day-by-day and gene-by-gene results are given in Table 1. Data from three independent biological replicates were analyzed with the Graph Pad Prism version 7.01 software.

**Table 1 cells-08-00583-t001:** Mitochondrial RNAs whose level increased (in black) or decreased (in purple) significantly in relation with the expression and mitochondrial import of the *orf77*-PKTLS RNA.

Conditions and Developmental Stage	Day 1	Day 2	Day 3	Day 4
Light *Early stage*		↓nad5a*		
		↓atp4**		
		↑rps7*		
		↑AOX1d*	↓AOX1d***	↓AOX1d***
Light *Intermediate stage*			↑cob**	↑rps7*
		↑nad5a**	↓ccmFN2***	↑rpl2**
	↑nad5b**	↑atp4*		↑rpl5*
	↓ccmFN2**	↑rpl2*		
		↑rpl5*		
			↑AOX1d***	↑AOX1d***
Light *Bolting stage*	↓nad2b**	↓atp4*	↑nad1b*	
	↑nad5a**	↓atp9****	↓nad2b*	
	↓nad5b**	↓rps4**	↓nad4**	
	↓nad9**	↓rps7****	↑nad5a***	
	↓cox1** ↓ccmB**	↓rpl2****	↓nad5b*	
	↓mttB*	↓rpl5**	↓nad6*	
	↓rpl5*	↓rrn18**	↓nad9**	
			↓cob***	
			↓cox1*	↓cox1****
			↓cox2**	↓atp9*
			↓cox3****	↓ccmFC***
			↑atp9*	↑mttB*
			↓ccmB***	↑rpl5**
			↓ccmFC*	↓rrn26*
			↓ccmFN1***	
			↓ccmFN2**	
			↓mttB**	
			↓rrn18***	
			↓rrn26**	

			↓AOX1a*	
	↓AOX1d**	↓AOX1d*	↓AOX1d*	
Dark	↑nad2b**	↑nad4*	↓nad1b***	↓ccmFC***
	↓nad5a**	↑nad5b*	↓nad2b**	↓matR*
	↑nad5b****	↑cox1*	↑nad5b**	↓mttB*
	↑nad9*	↓ccmFN2**	↓ccmFN2**	↓rrn18****
	↓cox2*		↓mttB**	↑rrn26**
	↓matR*		↓rrn26**	
	↑rps7**			
	↓rpl5*			
	↓rrn26*			

		↑AOX1d**		

Significant variations of nuclear RNAs coding for the mitochondrially targeted enzyme alternative oxidase (AOX1, underlined) are also included. Expression of the *orf77*-PKTLS transgene was induced at Day 0 and sampling was run every day until day 4 post-induction. Analyses were done by RT-qPCR with total RNA. Mitochondrial RNAs showing level variations versus control at Day 0 were not further considered. Data from three independent biological replicates were analyzed with the Graph Pad Prism software; * = *p* ≤ 0.05; ** = *p* ≤ 0.01; *** = *p* ≤ 0.001; **** = *p* ≤ 0.0001.

**Table 2 cells-08-00583-t002:** Selected *A. thaliana* nuclear-encoded RNAs upregulated upon knockdown of the mitochondrial *nad9* mRNA.

Modulation	Annotation / Function / Organellar Localization	Gene
Upregulated (opposite to *nad9*)	Basic helix-loop-helix (bHLH) DNA-binding superfamily protein; transcription factor; response to ethylene	AT1G05710
	ABI3-interacting protein 3, AIP3, PFD4, PREFOLDIN 4; protein chaperone; ABI3 is an auxin-inducible transcription factor	AT1G08780
	ATMYB60, Myb domain protein 60, MYB60; transcription factor; response to abscisic acid, jasmonic acid, salicylic acid	AT1G08810
	Agamous-like 87, AGL87, MADS-box family protein; transcription factor	AT1G22590
	GL2, GLABRA 2, HD-ZIP IV family of homeobox-leucine zipper protein with lipid-binding START domain; transcription factor	AT1G79840
	DREB subfamily A-6 of ERF/AP2 transcription factor family; one AP2 domain; ethylene-activated	AT2G22200
	AP2/B3-like transcriptional factor family protein; transcription factor	AT2G33720
	FMA (FAMA), basic helix-loop-helix (bHLH) DNA-binding superfamily protein; transcription factor/ transcriptional activator	AT3G24140
	EDF3, ethylene response DNA-binding factor 3; transcription factor; AP2 domain; ethylene responding	AT3G25730
	IAA30, indole-3-acetic acid inducible protein 30; transcription factor; response to auxin	AT3G62100
	B-BOX domain protein 23, BBX23; transcription factor	AT4G10240
	PUCHI, ethylene response factor (ERF) subfamily B-1 of ERF/AP2 transcription factor family; one AP2 domain; ethylene response	AT5G18560
	ARGOS, Auxin-regulated gene involved in organ size; response to ethylene, auxin; membrane, cytoplasm, mitochondrion	AT3G59900
	EMB3103, Embryo-defective 3103, PDM2, Pigment-Defective Mutant2; pentatricopeptide repeat (PPR) superfamily protein; endonuclease; chloroplast/mitochondrion	AT1G10910
	RNH1C, RNase H family protein, RNase H domain-containing protein; chloroplast	AT1G24090
	RING-finger, DEAD-like helicase, PHD and SNF2 domain-containing protein	AT2G40770
	AT-SR34B, Serine/arginine-rich protein splicing factor 34B, SR34B	AT4G02430
	Mitochondrial nuclease 1, MNU1; putative endonuclease or glycosyl hydrolase; mitochondrial RNA 5'-end processing; chloroplast/mitochondrion	AT5G64710
	EMB2394, Embryo-defective 2394; structural constituent of chloroplast ribosome; response to cytokinin	AT1G05190
	rRNA, cytosolic small ribosomal subunit	AT2G01010
	5.8S rRNA, cytosolic large ribosomal subunit	AT2G01020
	40S ribosomal protein S23 (RPS23A), ribosomal protein S12/S23 family protein; cytosolic small ribosomal subunit	AT3G09680
	Ribosomal protein L15, RPL15; large subunit of the chloroplast ribosome; response to cytokinin	AT3G25920
	EMB3126, Embryo-defective 3126, plastid ribosomal protein L1, PRPL1; ribosomal protein L1p/L10e family; chloroplast large ribosomal subunit	AT3G63490
	60S ribosomal protein L31, RPL31B; cytosolic large ribosomal subunit	AT4G26230
	U3 ribonucleoprotein, Utp family protein; rRNA processing	AT5G08600
	MA3 domain-containing translation regulatory factor 1, MRF1; colocalizes with cytosolic large ribosomal subunit; isomerase activity	AT5G63190
	snoRNA	AT1G03743; AT1G19373; AT1G19376; AT1G75166; AT2G35387; AT3G27865; AT3G47342; AT3G47347; AT3G58193; AT3G58196; AT4G39366; AT5G44286

Data taken from sparse PLS analysis of microarray results obtained with RNAs from light-grown *A. thaliana* plants at the bolting stage. Individual Day 0 to Day 4 RNA samples from three independent biological replicates were analyzed (i.e., 15 test samples and 15 control samples). Organelllar localization is mentioned where annotated. Selection was for genes of potential relevance in regulation mechanisms, including transcription factors, hormone signaling or RNA metabolism.

**Table 3 cells-08-00583-t003:** Selected *A. thaliana* nuclear-encoded RNAs downregulated upon knockdown of the mitochondrial *nad9* mRNA.

Modulation	Annotation/Function/Organellar Localization	Gene
Downregulated (as *nad9*)	MicroRNA414, mir414, short open reading frame 16, SORF16; miRNA; identified as a translated small open reading frame by ribosome profiling	AT1G67195
	B-BOX domain protein 25, BBX25, Salt tolerance homologue, STH; transcription factor; zinc ion binding	AT2G31380
	ATIBH1, IBH1, ILI1 binding BHLH 1 ILI1 binding bHLH 1; transcription factor; brassinosteroid signaling; gibberellic acid signaling	AT2G43060
	Arabidopsis thaliana response regulator 2, ARR5, ATRR2, IBC6, induced by cytokinin 6, response regulator 5, RR5; transcription repressor; cytokinin signaling	AT3G48100
	Basic helix-loop-helix (bHLH) DNA-binding superfamily protein; transcription factor	AT4G01460
	KELP; homodimers or heterodimers with the kiwi protein; transcriptional co-activator	AT4G10920
	WRKY DNA-binding protein 24, ATWRKY24, WRKY24; WRKY transcription factor group II-c	AT5G41570
	BOA, Brother of lux ARRHYTHMO; transcription factor; circadian clock; mRNA cell-to-cell mobile	AT5G59570
	SAUR53, small auxin-upregulated RNA 53, SAUR-like auxin-responsive protein family; mitochondrion	AT1G19840
	SAUR65, small auxin-upregulated RNA 65, SAUR-like auxin-responsive protein family; membrane; mitochondrion	AT1G29460
	ATCLE19, CLAVATA3/ESR-related 19, CLE19, embryo surrounding region 19, ESR19; receptor binding; signal transduction; mitochondrion	AT3G24225
	VQ motif-containing protein 29, VQ29; response to hypoxia	AT4G37710
	RNA-binding (RRM/RBD/RNP motifs) family protein	AT1G33470
	ATRNS1, Ribonuclease 1, RNS1; endoribonuclease	AT2G02990
	Eukaryotic translation initiation factor 2 (eIF-2) family protein; cytosol	AT1G76820
	EMB3113, Embryo-defective 3113, ribosomal protein S5, RPS5, SCA1, SCABRA 1; structural component of the 70S chloroplast ribosome; mitochondrial small ribosomal subunit	AT2G33800
	U3 containing 90S pre-ribosomal complex subunit	AT2G43110
	ATRAB8D, ATRABE1B, RAB GTPase homolog E1B, RABE1B; translation elongation factor; membrane; chloroplast	AT4G20360
	snoRNA	AT2G35382; AT2G43137; AT2G43138; AT2G43139; AT2G43141; AT4G02550; AT4G02555; AT4G13245

Data taken from sparse PLS analysis of microarray results obtained with RNAs from light-grown *A. thaliana* plants at the bolting stage. Individual Day 0 to Day 4 RNA samples from three independent biological replicates were analyzed (i.e., 15 test samples and 15 control samples). Organelllar localization is mentioned where annotated. Selection was for genes of potential relevance in regulation mechanisms, including transcription factors, hormone signaling or RNA metabolism.

**Table 4 cells-08-00583-t004:** Selected *A. thaliana* nuclear-encoded RNAs up- or downregulated upon knockdown of the mitochondrial *nad9* mRNA.

Modulation	Annotation/Function/Organellar Localization	Gene
Upregulated (opposite to *nad9*)	Small nuclear RNA U6acat, mRNA splicing	AT5G40395
Downregulated (as *nad9*)	ANAC028, NAC domain-containing protein 28, NAC028; transcription factor	AT1G65910
	Transcription elongation factor Spt5; KOW domain	AT2G34210
	MYR2, homeodomain-like superfamily protein; transcription factor	AT3G04030
	HVA22-like protein F, HVA22F; membrane protein; response to abscisic acid	AT2G42820
	ALY2 RNA-binding (RRM/RBD/RNP motifs) family protein; mRNA transport	AT5G02530

Data taken from sparse PLS analysis of microarray results obtained with RNAs from 10-day-old dark-grown *A. thaliana* seedlings. Individual Day 0 to Day 4 RNA samples from two independent biological replicates were analyzed (i.e., 10 test samples and 10 control samples). Selection was for genes of potential relevance in regulation mechanisms, including transcription factors, hormone signaling or RNA metabolism.

**Table 5 cells-08-00583-t005:** Selected *A. thaliana* nuclear RNAs up- or downregulated upon knockdown of the mitochondrial *nad9* mRNA and coding for proteins predicted to be mitochondrion-targeted.

Modulation	Annotation/Function/Organellar Localization	Gene
Upregulated (opposite to *nad9*)	auxin-responsive family protein (SAUR72)	AT3G12830
	auxin-responsive protein, putative (SAUR9)	AT4G36110
	auxin-responsive family protein (SAUR41)	AT1G16510
	auxin-responsive protein-related (SAUR77)	AT1G17345
	auxin-responsive family protein (SAUR53); mitochondrion	AT1G19840
	auxin-responsive family protein (SAUR71)	AT1G56150
	auxin-responsive family protein (SAUR52); mitochondrion	AT1G75590
	auxin-responsive protein-related (SAUR36); mitochondrion/nucleus (nucleus confirmed by GFP targeting experiments)	AT2G45210
	auxin-responsive family protein (SAUR59); mitochondrion	AT3G60690
	auxin-responsive protein, putative (SAUR25); mitochondrion	AT4G13790
	auxin-responsive family protein (SAUR1); chloroplast	AT4G34770
	auxin-responsive protein, putative (SAUR23)	AT5G18060
	MIF1 (MINI ZINC FINGER 1); transcription factor; response to abscisic acid, auxin, brassinosteroid, cytokinin, gibberellin; nucleus (cytosol reported from GFP data)	AT1G74660
	PLS (POLARIS), cytokinin and auxin responses; mitochondrion	AT4G39403
	Tetratricopeptide repeat (TPR)-like superfamily protein; mitochondrion	AT1G28690
	S-RBP11, SMALL RNA-BINDING PROTEIN 11; salt stress response; chloroplast	AT5G06210
Downregulated (as *nad9*)	ARGAH2, Arginine amidohydrolase 2, response to jasmonate; chloroplast/mitochondrion (confirmed by MS data)	AT4G08870
	Tetratricopeptide repeat (TPR)-like superfamily protein; chloroplast/mitochondrion (chloroplast confirmed by MS data)	AT2G37230
	pentatricopeptide (PPR) repeat-containing protein	AT3G62470
	pentatricopeptide (PPR) repeat-containing protein	AT4G01030
	EMB1417 (embryo-defective 1417), PPR protein; RNA binding; endonuclease activity	AT4G21190
	pentatricopeptide (PPR) repeat-containing protein	AT4G21880
	pentatricopeptide (PPR) repeat-containing protein	AT5G65560
	EMB1586, Embryo-defective 1586, increased size exclusion limit 1, ISE1; DEAD-box RNA helicase; chloroplast/mitochondrion (mitochondrion confirmed by GFP data)	AT1G12770
	AGS1, AHG2-1 suppressor 1, bacterial-type poly(A) polymerase; mRNA polyadenylation; chloroplast/mitochondrion	AT2G17580
	ATTRM2A, TRM2A, tRNA methyltransferase 2A, RNA methyltransferase family protein; cytoplasm	AT3G21300
	ATP-dependent RNA helicase	AT5G39840
	ARFB1A, ATARFB1A (ADP-ribosylation factor B1A); GTP binding; protein transport; Golgi apparatus	AT2G15310

Data taken from microarray analysis of RNAs from 10-day-old dark-grown *A. thaliana* seedlings. Individual Day 0 to Day 4 RNA samples from two independent biological replicates were analyzed (*i.e.,* 10 test samples and 10 control samples). All proteins included have a prediction consensus for mitochondrial targeting in the SUBA database [44] (http://suba.plantenergy.uwa.edu.au/). Experimentally reported localization compiled in the SUBA database is also indicated. Selection was for genes of potential relevance in regulation mechanisms, including hormone signaling or RNA metabolism.

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
