# Peer review of "Mitochondrial Transcriptome Control and Intercompartment Cross-Talk During Plant Development"

_cells, 2019, doi:10.3390/cells8060583_

Round 1

Reviewer 1 Report

The manuscript „Mitochondrial Transcriptome Control and Inter-Compartment Cross-Talk During  Plant Development” presents the analysis of mitochondria-nucleus transcriptomic coordination, in the plants, at different ontogenetic stages of development. The functional  coordination within mitochondria and between mitochondria an nucleus is under investigation for some time and many open questions are still present. The developmentally oriented research presented in the manuscript is in that regard of big value and therefore fits well with the Journals profile. In general I find the experimental set-up, and results as a whole, interesting. However in my opinion certain issues need to be improved to enable appreciation of the data and drawing clear-cut conclusions, in the context of the ontogenetic changes occurring in plants. I also feel that some of the obtained results can be more thoroughly discussed from the developmental perspective. Therefore, the manuscript in my opinion needs a revision.

Major concerns:

1.      Few concern about the results, and the way they are presented, from Figure 2 and 3:

-        The authors state that no phenotypes after induction (on all developmental stages) were generally observed. The way the plants are presented makes it difficult to me to assess that unambiguously. First of all only selected lines are shown. Personally I think that when no phenotypes were ever observed authors can show the phenotypes of only one line (and all its developmental stages), but in the text there is a need for a clear statement: that for the simplicity authors present only pictures related to one line, as for all the situation was the same. Secondly, the “no-phenotype” statement is quite strong, as the phenotypes can be too subtle to be seen on such pictures (as indeed 4 days may be too short to develop drastic changes). Therefore my question is: did the authors analyse the plants in more detail, for example made pictures of individual leaves for phenotype comparison? If present, such subtle phenotypes could be added to supplements.

-        I think it would be better to make one figure from Figure 2 and 3 (with the phenotype of only one selected line in all developmental stages).

-        The following naming is a bit confusing for me: 

a)     flowering initiation stage vs late stage: I would like to additionally know how high the inflorescence stem was (or how many open flowers were present on the stem) at these two stages. Number of leaves is not very informative here. I was also wondering what plant organs were taken for analyses in that two flowering stages: whole plants or just leaf rosette.

b)     what does it mean non-mature and mature leaves? How authors judged the end of maturation process within the leaves. Or do the authors mean juvenile vs  adult (but then at early stage also first adult leaves are already appearing, as I assume from the pictures).

-        What worries me is that I think that, in general, certain important results are lacking.

a)     In my opinion authors have to show graphs from all developmental stages for all the analysed genes. I don’t understand why in terms of atp9, cox3 and sdh3 only selected stages are presented (and why in terms of cox3 and sdh3 the data about the level of induction is not presented). I assume authors have such data, and it is a valuable information if one is to draw conclusions in the developmental context. When all such data would be presented in one Figure it would strongly prove the point authors make from their results. Even if something has been already previously published, it was in a different research context.

b)     Importantly, for proper concluding authors should provide data comparing the transcript levels of the endogenous genes (nad9, atp9, cox3 and sdh3) between the developmental stages (without induction). Maybe the different strength of endogenous genes transcription reduction upon estradiol induction in subsequent developmental stages is an effect of different level of these genes expression at a particular developmental stage. Without such data we cannot rule out such a possibility.

Such improvements would allow to draw stronger and more clear-cut conclusions.

-        I generally lack the information about the statistical significance of the differences on the graphs

2.      I think that the Suppl Table 1 should be a regular table within the manuscript proper, as not only the number of genes is of relevance but also how particular genes up/down regulation overlap between developmental stages. I would appreciate if authors would include such description (also regarding  the differences in the strength of up/down regulation of a given gene between developmental stages) also in the text of the section Results and/or Discussion.

3.      In terms of the tables:

-        the description at the beginning of the tables starting with: “Selected A. thaliana ……”, should include a short information on what basis the selection was made.

-        Table 4 and 5 can be presented as one table

4.      Did the authors confirm/validate their microarray data with use of qRT-PCR (performed on few selected genes)?

-        I lack a short information why authors chose for microarray experiment the fourth day upon induction

Minor remarks about the problematic sentences, which in my opinion need improvement to be properly understood:

5.      From 48: “Strikingly, studies on the regulation of mammalian mitochondrial gene expression essentially pointed out complex post-transcriptional processes [3,4], including enzymatic nucleotide modification of organellar transcripts and coordination of mitochondrial and cytosolic translation [5,6]. “ – please improve

6.      From 752: “Our results support the idea that a fast 752 intergenomic communication, rather than signals that would be generated by metabolic 753 dysfunction, underlies these processes.” – please improve

7.      From 64: ”…..mitochondrial biogenesis and functioning involve both nuclear 64 gene expression and mitochondrial gene expression. The protein complexes of the electron transport 65 chain (ETC) are assembled from nucleus-encoded subunits and mitochondrion-encoded subunits,…” – can be rewritten to avoid repetitions of the same words.

8.      From 88: “These different regulation 88 processes enable the adjustment of mitochondrial…” – what are “regulation processes”?

9.      From 97: “Experiments were based on the analysis and manipulation of the transcriptome in Arabidopsis thaliana through a novel approach set up in our laboratory. Establishment of the plant transcriptome has been documented in more detail along seed germination in A. thaliana [29].” – it is not fully clear from such description what was analysed: mitochondrial or nuclear transcripts, or the coordination?

10.   From 122: “..previously [Val] an alternative..” – please provide a proper citation

11.   I think a sentence or two describing the function of ATP9 gene and why it is important to study that particular gene in the paragraph starting from 135 would be a nice addition

12.   From 144: “…can functionally compete with…” – what exactly authors mean by that?

Final conclusions:

This kind of research, where developmental perspective was taken into consideration, is of big value and has a great potential, but I think it should be improved prior to publication.

Author Response

Please see the uploaded pdf file "Answers to Reviewer 1

Reviewer 2 Report

I suggest publication, pending revisions. The most important issue I found is related to sampling, biological replicates, and sample size. The manuscript “Mitochondrial Transcriptome Control and Inter-Compartment Cross-Talk During Plant Development” by Niazi et al., presents work on the important and understudied area of mitochondrial genetic regulation and the retrograde transcriptional coordination. The manuscript is well written. The topic is of interest, and builds on previous work published by this group. Overall, the data presented is solid, experimental procedures are presented in sufficient detail, and all sections were adequately developed. The manuscript could be further improved by considering the following suggestions: - The Introduction is well documented, but could be a little more concise, in particular the last two paragraphs, which could be reorganized as part of the Discussion. - In the Materials and Methods section, it is not clear which plant parts were used, what was the sample size (how many plants were sampled for each biological replicate), and how many biological replicates were analyzed. Is there expectation that all tested genes have similar expression profiles in roots and leaves for example? If this was done using entire plants, it should be spelled out. - The bouncing-back effect shown in Figs. 2 and 3 is intriguing. In particular for these experiments it would be important to run appropriate controls, such as samples from non-transformed plants collected at the same time-points as the transgenic ones, to reflect the same developmental stages. - When first introduced, acronyms should be spelled out (e.g. TIM and TOM in line 105) - In line 122, replace [Val] with the appropriate reference. Please check out all references cited afterwards to ensure that there are no numbering errors after this one.

Author Response

Please see the uploaded pdf file "Answers to Reviewer 2"

Reviewer 3 Report

This manuscript addresses the regulatory mechanism of mitochondrial transcriptome and proposes transcriptional coordination mechanisms within mitochondria and between mitochondria and nucleus. In this manuscript, directed manipulation of mitochondrial mRNA is established. This method seems to have originality. However, there are several concerns regarding the interpretation of data. Thus, I cannot recommend the publication of this manuscript in “Cells” journal.

1. From the results of figure 2 and 3, this manuscript concludes that the homeostasis of mitochondrial transcriptome is depend of growth stage. However, there is the possibility that the expression level of transgene is altered due to the difference of sensitivity to estradiol in each growth stage. The authors should show that the expression levels of transgene after estradiol treatment in all growth stage is in same level. In addition, there are concerns regarding this experiment as listed below;

- Does not control treatment affect the mRNA level of targeted genes? The effect of submergence to liquid medium should be investigated.

- Does not estradiol treatment affect the mRNA level of targeted genes in control plants, such as non-transformant or pER8 vector control? The side effect of estradiol should be investigated.

- The concentration and dose-dependency of estradiol should be shown.

- The data in figure 2 and 3 is presented in relative level. However, the absolute copy number should be investigated, because the ratio of copy numbers between transgene and targeted gene probably affect the efficiency of knock-down.

2. I cannot follow the reason why CMS RNA-triggered modulation is used for mitochondrial transcriptome analyses, although the description in lines 452-454. In the manuscript, ribozyme-mediated knockdown is used for nucleus transcriptome analyses. The authors should show the same phenomenon with figure 5 is appears in ribozyme-mediated knockdown plants.

Author Response

Please see the uploaded pdf file "Answers to Reviewer 3"

Reviewer 4 Report

In this manuscript mitochondrial transcriptome of plants has been investigated during plant development using two different strategies. In both strategies, the general idea was to disturb the mitochondrial transcriptome at different stages of plant development for then investigating the induced regulatory impact and signaling responses at the transcriptome level. The final message resulting from such studies support the idea that mitochondrial RNA levels are controlled and coordinated at the early and young stages of plat development and that interferring with specific mitochondrial transcripts cause a retrograde response that affects the nuclear transcriptome and triggers an anterograde response. The methodological strategies used to disturb mitochondrial transcriptome are well designed and very interesting. However, the outcomes deriving from them are not so informative since they do not provide information on how the coordination and cross-talk between the cellular compartments occur. Therefore, the obtained results provide just a general overview on a high complexity of cellular coordination mechanisms, which still remain quite nebulous. In conclusion, my view on this manuscript is partially positive and I think that the authors should at least improve it in terms of readibility by  better describing what has been accomplished and what still need to be understood. In the actual form the message of the manuscript is quite diluted, dispersive and not well focused. I recommend to shorten the manuscript and improve the Discussion section. 

Author Response

Please see the uploaded pdf file "Answers to Reviewer 4"

Round 2

Reviewer 1 Report

Manuscript: Cells-479040 

Mitochondrial transcriptome control and inter-compartment cross-talk during plant development 

The manuscript in my opinion has been improved and clarified, thus, can be now considered for publication.

Reviewer 3 Report

The revised manuscript have been addressed the issues raised in previous review. The manuscript is now suitable for publication.